# Improvement of the Performance of Chitosan—*Aloe vera* Coatings by Adding Beeswax on Postharvest Quality of Mango Fruit

**DOI:** 10.3390/foods10102240

**Published:** 2021-09-22

**Authors:** Usman Amin, Muhammad Kashif Iqbal Khan, Muhammad Usman Khan, Muhammad Ehtasham Akram, Mirian Pateiro, José M. Lorenzo, Abid Aslam Maan

**Affiliations:** 1National Institute of Food Science and Technology, University of Agriculture, Faisalabad 38000, Punjab, Pakistan; uamin2@ncsu.edu (U.A.); Kashif.khan@uaf.edu.pk (M.K.I.K.); 2Department of Food Engineering, University of Agriculture, Faisalabad 38000, Punjab, Pakistan; ehtasham.akram@uaf.edu.pk; 3Department of Energy Systems Engineering, University of Agriculture, Faisalabad 38000, Punjab, Pakistan; usman.khan@uaf.edu.pk; 4Centro Tecnológico de la Carne de Galicia, Avd. Galicia No. 4, Parque Tecnológico de Galicia, San Cibrao das Viñas, 32900 Ourense, Spain; jmlorenzo@ceteca.net; 5Área de Tecnología de los Alimentos, Facultad de Ciencias de Ourense, Universidad de Vigo, 32004 Ourense, Spain

**Keywords:** mango, bioactive, coatings, biodegradable, *Aloe vera*, chitosan

## Abstract

The effect of the application of chitosan–*Aloe vera* coatings emulsified with beeswax (0, 0.5, 1, 1.5 and 2%) during storage of *Mangifera indica* L. (cv Anwar Ratol) was investigated. Particle size of emulsions was reduced significantly with an increase in beeswax concentration. Water vapor permeability of the coatings was reduced by 43.7% with an increase in concentration of beeswax to 2%. The coated mangoes (at all concentrations of beeswax) exhibited reduced weight loss, delayed firmness loss, minimized pH change, maintained the total soluble solid contents, and retained free radical scavenging activity and total phenolic contents when stored at 18 °C and 75 ± 5% R.H. The best results were produced with a formulation containing 2.0% beeswax. Antimicrobial properties of chitosan and *Aloe vera* coatings were also improved with an increase in beeswax concentration and remarkably reduced the disease incidence in mangoes. In conclusion, beeswax-emulsified chitosan–*Aloe vera* coatings can be effectively used to increase the shelf life and marketable period of mangoes.

## 1. Introduction

Mango (*Mangifera indica* L.), commonly known as the “king of fruits”, is the most popular fruit in the world due to its attractive color, taste, nutritional value and health benefiting impact [1]. After citrus, mango is the second largest fruit crop in Pakistan, making it the fifth largest producer among 87 countries. This fruit provides a significant socio-economic contribution to the economy [2,3]. Several postharvest changes i.e., continuous respiration, disease incidence, weight loss etc., end in loss of quality of mango fruits [4]. Various strategies have been developed to reduce waste. Among them, physical control (i.e., controlled atmospheric (CA) storage), chemical control (i.e., chemical preservatives) and fungicides have been used as a common practice to extend the shelf life of fruits and vegetables [5]. Changes in relative humidity, gas composition and temperature fluctuations during CA storage result in loss of weight, fruit texture, chlorophyll decay, respiration and microbial infection. Similarly, chemical preservatives pose a threat to the environment and human health. Thus, there is a need to develop human environment friendly alternates that may extend the shelf life of fruits. 

In this regards, biodegradable coatings have been considered as an assistive approach to increase the shelf life and acceptability of stored fruits [6]. The term “biodegradable” refers to the condition that these coatings are prepared from natural polymers and can be decomposed by microorganisms after consumption of fruits [7]. Thus, they are safe, sustainable and environment friendly tool to assist the preservation of fruits and vegetables [8]. They have also been used as a carrier to several functional ingredients i.e., antioxidants and antimicrobial agents, to prevent the growth of microorganisms and oxidation reactions in fruits and vegetables [9,10]. These types of coatings are commonly known as active coatings. Performance of biodegradable coatings depends on type of polymer, bioactive agent, and lipid used, as lipids play an essential role to prevent moisture loss and preserve the freshness of the food product. In the past, scientists focused only on the development of these types of coatings. Therefore, the literature reports limited application of these coatings to enhance the shelf life of food products.

Chitosan, an amino-polysaccharide, is a non-toxic, biodegradable, edible and excellent film forming material [11]. Positively charged amino groups of chitosan interact with negatively charged cell walls of microorganisms, exhibiting antimicrobial potentials of chitosan [12]. Incorporation of *Aloe vera* in combination with chitosan has been reportedly used in the literature to improve the functional properties of biodegradable coatings [9]. Combination of chitosan and *Aloe vera* has been used to enhance the shelf life and disease control in cucumber [13], blueberries [14], strawberries [15], papaya [16], fresh-cut kiwifruit slices [17], fresh-cut red bell pepper [18] and several other applications.

To further improve the barrier properties and functionality of biodegradable coatings, beeswax has been reportedly used in coating solutions. Previously, beeswax coatings have been applied on several fruits such as kinnow [19], guava [20], blackberries [21], sapodilla fruits [12] and other similar fruits. These coatings had successfully minimized weight loss and maintained total soluble solids, skin color, and bioactive components i.e., vitamins, antioxidants, flavonoids.

Chitosan–*Aloe vera* coatings and beeswax has been separately evaluated on various fruits, however, no application of Chitosan–*Aloe vera* coatings emulsified with beeswax have been reported on any of the fruits and vegetables, to the best of our knowledge. Moreover, authors previously developed edible films using different concentrations of these materials [22]. Films revealed excellent physicochemical and mechanical properties, which strengthened their potential for food packaging application. Therefore, the purpose of current study was the development and characterization of beeswax-emulsified chitosan–*Aloe vera* coatings. Furthermore, effects of application of these coatings on physicochemical, mechanical and functional properties of mango fruits have been studied.

## 2. Materials and Methods

### 2.1. Selection of Healthy Mango Fruits

Fully ripened mango (*Magnifera Indica* L. cv. Anwar Ratol) fruits were collected from the local market of Faisalabad, Pakistan. Fruits with uniform shape, color, size were selected, while diseased or blemished fruits were discarded. Sorted fruits were first washed with distilled water and then dipped in NaOCl (3%) solution for 3 min to sterilize the mango fruits. Afterward, fruits were washed and cleaned with soft tissues to remove small water droplets. Afterward, fruits were randomly distributed into five different groups (each containing 20 fruits) prior to treatments. Each fruit was assigned a specific name (Control*_i_*, S1*_i_*, S2*_i_*, S3*_i_*, S4*_i_*; where *i* = 1, 2, 3, 4, … n) and stored at 8 °C prior to coating. Uncoated samples were considered as control samples. 

### 2.2. Procurement of Coating Material

Chitosan (origin: crab shells), beeswax (yellow soft), gallic acid (MW: 170.12), Folin-Ciocalteu (2 N), glycerol (MW: 92.09), tween 20 (conductivity: 50 µS/cm), acetic acid (≥99.5%) and NaOCl (4–4.99% available chlorine) were purchased form Sigma Aldrich, USA. Fresh *Aloe vera* plants were harvested from a local nursery in Faisalabad, Pakistan. 

### 2.3. Preparation of Coating Solutions 

Preparation of coatings was performed in three steps. First, chitosan solution was prepared by slowly dissolving 1 g of chitosan in 100 mL of 2% acetic acid solution (*v*/*v*) on a magnetic stirrer (350 rpm) to get 1% chitosan solution (*w*/*v*) [23]. For *Aloe vera* gel preparation, *Aloe vera* leaves were peeled with stainless steel knife and gel was extracted. It was then blended and homogenized in a blender and filtered to remove impurities and debris material. A 20% *Aloe vera*–chitosan solution was prepared by mixing 20 mL of *Aloe vera* with 80 mL of prepared chitosan solution (1%). This combination was selected because films of these materials revealed superior barrier and mechanical strength in our previous study [22]. Afterward, glycerol (0.2 g) and tween 20 (0.2 g) were added in chitosan–*Aloe vera* solution. Finally, beeswax was melted at 70 °C and mixed with the chitosan–*Aloe vera* solution in the proportions described in Table 1. Coating formulations were homogenized at 13,500 rpm for 3–4 min using ultra-turrax homogenizer (IKA-T18-Germany).

### 2.4. Application of Coatings on Mango Fruits

Coatings (slightly yellowish in color) were applied on mango fruits by dipping the mango fruits in the coating solution. Mango fruits in each group were dipped in the coating formulations, separately, for the same time under ambient conditions (20 °C, R.H. 70%). After drying of coatings, all the samples were stored at 18 °C and 75 ± 5% R.H. These conditions were selected to mimic the market environment, as this study attempted to increase the marketable life of mangoes. For subsequent physicochemical analyses, each sample was marked with the group name and fruit number for an effective storage study (e.g., sample S1 as S1_1_, S1_2_, S1_3_, etc.). Then, each sample was evaluated for physicochemical characteristics in 7 day intervals for three weeks.

### 2.5. Characterization of Coating Emulsions

#### 2.5.1. Particle Size Analysis

Prepared emulsions were subjected to particle size analysis using particle size analyzer (Bettersizer ST, Bettersize Instruments Ltd., Dandong, Liaoning, China).

#### 2.5.2. Emulsion Stability

Stability of the emulsions was determined by following the method reported by Amin et al. [22] with little modifications. Briefly, 6 mL from each coating formulation was taken in a test tube on test tube rack and placed under ambient condition for three days. Stability of the emulsion was measured by the following equation:(1)$$\mathrm{Emulsion}\;\mathrm{stability}=\frac{h_{o}-h_{t}}{h_{o}}$$
where *h_o_* and *h_t_* is the initial and final height of the emulsion in the test tube after 3 days’ time, respectively.

#### 2.5.3. Water Vapor Transmission Rate

Water vapor permeability of the coatings was measured according to ASTM method [24] with few modifications. Coating solutions were casted into films and dried at 23 ± 2 °C on a smooth horizontal surface until the coating solutions shaped into a thin sheet or films. Glass tubes (5 × 10 cm) were filled with desiccant (silica gel) and dried films were fixed over the face of the tube (by using gel). Films were placed in the climate chamber (POL-EKO-APARATURA-KK-350, Poland) and maintained at 38 ± 0.6 °C and 90 ± 2% R.H. Change in weight of tubes was measured periodically for 7 days to obtain moisture transfer rate. Water vapor permeability (WVP) was calculated by the following equation:(2)$$\mathrm{WVP}=\frac{∆m\times T}{A\times t\times ∆P}$$

Here, ∆*m* is the change in mass (g) before and after the specific time, *A* is the area (m^2^), *t* is the time (hours), *T* is the thickness of films and ∆*P* is the difference in pressure at saturated pressure and pressure under the testing conditions. 

#### 2.5.4. Diffusion Coefficient

The moisture loss in term of diffusion through the coatings was determined by using Fick’s law of diffusion, written as:(3)$$MR=\frac{M_{t}-M_{e}}{M_{o}-M_{e}}=\frac{8}{\pi^{2}}exp\left(\frac{-\pi^{2}Dt}{4L^{2}}\right)$$

Here, *MR*, *D*, *t* and *L* are the moisture ratio, effective diffusion coefficient (m^2^/s), time (s) and thickness (m) of the sample layer distributed in the drying chamber, respectively. Moreover, *M_t_*, *M_e_*, *M_o_* represents moisture levels at time interval of *t*, initial and equilibrium, respectively. The diffusion coefficient can be calculated from the slope of line (*α*) drawn between ln(*MR*) and time as:(4)$$\alpha =\left(\frac{\pi^{2}Dt}{4L^{2}}\right)$$

### 2.6. Physicochemical Analysis of Coated and Uncoated Mango Fruits

#### 2.6.1. Percentage Weight Loss

Percentage weight loss of fruits was measured by using digital weight balance. The weight of each marked sample was measured before the start of experiment. Change in weight of stored samples was observed and percentage weight loss was calculated by the following equation [19]:(5)$$\mathrm{Weight}\;\mathrm{loss}\;\left(\%\right)=\frac{W_{i}-W_{f}}{W_{i}}$$
where *W_i_* is the initial weight of the coated fruit and *W_f_* is the weight after specific time interval during storage.

#### 2.6.2. Firmness

Firmness of mangoes was measured by using a Texture Analyzer (TA XT Plus, Stable Micro Systems, Godalming, Surrey, UK) as performed by Meindrawan et al. [25] with slight modification in speed and replications. Three samples from each group of S1, S2, S3, and S4 were selected randomly after 0, 7, 14, 21 days and repeatedly placed on the stage of Texture analyzer. Sampling probe was inserted inside the fruit sample at a speed of 3 mm/s and repeated three times on different areas of a sample. The penetration force was expressed in grams the fruits can bear.

#### 2.6.3. Total Soluble Solids

The total soluble solid contents of the fruits were determined using digital refractometer (DR201-95, KRÜSS Optronic GmbH, Hamburg, Germany). The pulp of the mangoes was extracted and homogenized for uniform distribution of solid particles in the solution. A drop from the pulp was placed on the sample stage of the refractometer, and the results were expressed as °Brix.

#### 2.6.4. pH

Homogenized fruit pulp was subjected to pH measurement using pH meter (ST5000, OHAUS, Parsippany, NJ, USA) through the method described by Eshetu et al. [26]. pH meter was first calibrated, and the probe was inserted into the homogenized sample. The value of pH was noted as the concentration of hydronium ion on pH scale 0–14. 

#### 2.6.5. Total Phenolic Contents (TPC)

Total phenolic contents were measured by the Folin-Cicalteau method [8]. Briefly, 0.3 mL of the extract prepared for antioxidant activity was mixed with 1.2 mL of 7% sodium carbonate solution. The mixture was then mixed with 1.5 mL of Folin reagent and placed on shaker for 1.5 to 2 h. The absorbance was measured at 765 nm by UV/VIS spectrophotometer (T80). Gallic acid calibration curve (10 ppm–1000 ppm) was used to measure TPC, and results were expressed as mg gallic acid (GAE)/100g MP.

#### 2.6.6. Total Antioxidant Capacity 

The antioxidant activity was assayed by the method adopted by Ebrahimi et al. [8]. To prepare the extract, homogenized fruit pulp was added in 80% methanol and placed on the shaker for 3–4 h. After this, DPPH solution was prepared by adding 0.025 g of DPPH in 10 mL of 85% ethanol. A volume of 50 μL of the prepared extract was mixed well with 950 μL of DPPH solution. The samples were then placed in the dark room for 30 min. Absorbance by the samples was measured at 517 nm using UV/VIS spectrophotometer (T80). Free radical scavenging activity (%) was measured by using the following equation:(6)$$\mathrm{Free}\;\mathrm{redical}\;\mathrm{Scavenging}\;\mathrm{activity}\;\left(\%\right)=\frac{{\left(\mathrm{Absorbance}\right)}_{\mathrm{DPPH}}-{\left(\mathrm{absorbance}\right)}_{\mathrm{Sample}}}{{\left(\mathrm{Absorbance}\right)}_{\mathrm{DPPH}}}$$

#### 2.6.7. Decay Incidence

Decay incidence on mango fruits was assessed through the method reported by Khaliq et al. [27] and Eshetu et al. [26]. Fruits were carefully observed for fungal and bacterial infection after 7, 14 and 21 days. Scale was observed as 1 = no decay, 2 = 0–5% decay, 3 = 5–25%, 4 = 25–50%, 5 = 50–75%, 6 = >75%. Percentage of fungal and bacterial decay was calculated by the following equation:(7)$$\mathrm{Decay}\;\mathrm{incidence}=\frac{\sum \left(\mathrm{decay}\;\mathrm{level}\;\times \;\mathrm{Number}\;\mathrm{of}\;\mathrm{fruit}\;\mathrm{at}\;\mathrm{that}\;\mathrm{level}\right)}{\mathrm{Total}\;\mathrm{number}\;\mathrm{of}\;\mathrm{fruits}\;\times \mathrm{Maximum}\;\mathrm{decay}\;\mathrm{level}}\times 100$$

### 2.7. Statistical Analysis

Data was obtained in triplicate and subjected to statistical analysis at 5% confidence interval. Characteristics of emulsion were analyzed through ANOVA and differences between the means was analyzed through pairwise comparison using SAS software package (SAS institute, Cary, NC, USA). Shelf life study parameters were analyzed using MATLAB curve fitting tool (MATLAB 2016a, Natick, MA, USA).

## 3. Results and Discussions 

### 3.1. Characterization of Emulsion Coatings

#### 3.1.1. Particle Size and Emulsion Stability

Particle size and size distribution are the most important parameters of emulsions, determining their characteristics such as rheology, appearance, stability, etc. [28]. The effect of beeswax concentration on logarithm distribution of particles in emulsions is shown in Figure 1. A non-significant difference was observed in particle size distribution with increase in concentration of beeswax in chitosan—*Aloe vera* emulsions. The average particle diameter for all the emulsions was observed to be around 4 µm. Xie et al. [29] observed similar behavior for emulsions of beeswax in carboxymethyl chitosan and cellulose nanofibrils. Contraction in particle size range with increased beeswax concentration might have produced more shear during homogenization and produced more rod-like structures during hardening of beeswax upon cooling [30]. Moreover, emulsions remained stable for all concentrations of beeswax after 72 h, without any separation of dispersed phase. This reveals better dispersibility of beeswax in chitosan—*Aloe vera* emulsions.

#### 3.1.2. Water Vapor Permeability and Diffusivity

Results for water vapor permeability of films is shown in Figure 2. The highest value of WVP was observed for 0.5% beeswax and the lowest for 2% concentration of beeswax. Thus, increasing the concentration of beeswax from 0.5 to 2% resulted in 43.78% reduction in water vapor permeability of films. This is obvious due to increase concentration of wax in the coatings. Pérez-Vergara et al. [31] attributed the reduction in moisture transfer to a large number of long-chain fatty alcohols and alkanes present in the orthorhombic structure of beeswax. Zhang et al. [32] have reported that synergistic interaction between chitosan and beeswax films exhibits better water barrier properties compared to pure beeswax films.

Similarly, diffusion of moisture through the films was also calculated and drawn in Figure 2. Results of diffusion coefficient revealed that an increased concentration of beeswax in *Aloe vera* and chitosan blend had significantly reduced moisture diffusion through the coatings. Initially, moisture diffusion for uncoated mango fruits was observed to be 1.5 × 10^–10^ m^2^/s, which was reduced to 7.94 × 10^–11^ m^2^/s as the concentration of beeswax increased to 2% in the coating formulations.

### 3.2. Physicochemical Characteristics of Coated and Uncoated Mangoes

#### 3.2.1. Weight Loss

Fruits and vegetables are mainly composed of water, which is known as the indicator of freshness of fruits and vegetables. Fluctuation in storage temperature and relative humidity of the storage environment may trigger respiration and transpiration rate, which may cause weight loss in fruits and vegetables [26]. In the current study, weight loss of mangoes was significantly reduced by all coating solutions as shown in Figure 3a. However, weight loss was observed to increase throughout storage. Mangoes coated with treatment S4 (containing 2% beeswax) exhibited minimum weight loss during storage periods. Increased concentration of beeswax increased the hydrophobicity of coatings, which conversely reduced the moisture removal, and consequently, the weight loss from the fruits. This reduction of weight loss in polysaccharide coatings may be associated to better crosslinking between hydroxyl group of polysaccharide and hydrophilic substance (such as phenols) in coatings through hydrogen bonding [33]. The results are supported by the findings of Nasrin et al. [34], who used beeswax coatings in combination with coconut oil and observed the similar behavior of weight loss in lemons. Eshetu et al. [26] also reported that 2% beeswax in chitosan coatings preserved moisture inside the mangoes.

#### 3.2.2. Total Soluble Solids

Figure 3b exhibits the effect of beeswax concentration and storage time on total soluble contents of mango fruits. Concentration of beeswax in coating solutions significantly affected total soluble contents in mango fruits. A rapid change in degree brix was observed in the control sample, which increased from 60.08 to 65.13 °Brix during storage time. However, a non-significant change in brix was observed for coated fruits. This may be due to the reduced conversion of acid into sugar during 21 days of storage. Chitosan–*Aloe vera* coatings with 2% beeswax revealed minimum change in TSS (from 60.08 to 61.74 °Brix). Moalemiyan et al. [35] reported that an increase in TSS may be associated with weight loss causing an increase in sugar concentration. Moreover, conversion of complex carbohydrates present in the mangoes into soluble sugars also increased the TSS contents. Coatings provide an efficient cover from the environmental oxygen and hinder the metabolic activities responsible for rapid conversions of acids into sugar [19]. As explained earlier, a slight change in weight loss was observed for coatings containing 2% beeswax, which may be responsible for a slight increase in TSS contents in mangoes treated with S4.

#### 3.2.3. pH

Concentration of beeswax in the coating solution significantly affected the pH of mangoes during storage, as shown in Figure 3c. A linear and rapid increase in concentration of hydronium ion was observed for uncoated fruits. As the storage time was increased, their change in pH became more pronounced. All the coating formulations significantly hindered the change in pH as compared to the uncoated treatments. However, treatment S4 was the most efficient in reducing the rate of change in pH. Increase in pH values is attributed to an increase in conversion of organic acids into sugars [35]. As explained in the section above, TSS values changed slightly for mango coated with 2% beeswax (S4). Therefore, minimum variations in pH were observed for samples treated with S4 as compared to control samples. Findings of Velickova et al. [23] support the results obtained in the current study, who applied chitosan–beeswax coatings on strawberries. 

#### 3.2.4. Firmness

Firmness of fruit is an important indicator of freshness and quality. Firmness of coated mango fruits was tested and compared with uncoated mangoes. Figure 3d reveals the effect of beeswax concentration and storage time on firmness of mango fruits. Firmness of mango fruits was observed to decrease linearly with an increase in storage time for all coating treatments. However, the rate of firmness loss was consistently reduced with addition of beeswax up to 2%. Application of hydroxypropyl and carnauba wax coatings [36] also revealed similar results. Klangmuang and Sothornvit [37] reported that loss in firmness of mango fruits is attributed to cell wall digestion by different enzymes during ripening. Enzymes such as pectin esterase and polyglacturonase trigger the hydrolysis of starch and pectin present in the cell wall. This causes the loss in cellular turgor and subsequent loss of firmness of mango fruits. Emulsified coatings slowed down the metabolic activity in the fruits by controlling the environment of coated fruits, which maintained the firmness in coated samples [38].

#### 3.2.5. Free Radical Scavenging Activity

Antioxidant activity of mangoes has been attributed to a number of compounds, i.e., organic acids, chlorophyl, phenolic compounds etc. The role of coatings in maintaining antioxidant activity during storage is shown in Figure 4a. Uncoated samples exhibited maximum reduction in antioxidant activity (84.89% to 80.66%) during the storage of 21 days. The loss of antioxidant activity was reduced with an increase in beeswax concentration, with minimum reduction observed in S4. Reduction in free radical scavenging activity was more pronounced at the end of the first week for all samples. These results are in correspondence with Ebrahimi et al. [8], who applied guar gum–*Aloe vera* coatings containing *Spirulina platensis* on mango fruits. 

#### 3.2.6. Total Phenolic Contents (TPC)

Phenolic compounds are present in almost all plants as essential secondary metabolites. They also provide protective mechanisms against different pathogens. However, phenolic compounds start to decrease with an increase in storage time in fruits and vegetables [27]. The effect of coatings on phenolic contents of coated and uncoated mangoes during storage is presented in Figure 4b. All the coating formulations were observed to efficiently retain the TPC throughout the storage period, with S4 being the most efficient. TPC in the control sample reduced from 0.342 to 0.3201 mg GAE/g of mango pulp. Maintenance of phenolic compounds with increase in beeswax concentration can be attributed to the low oxygen permeability. Hinderance in oxygen availability to mango fruits through coatings slowed down the enzymatic degradation and oxidation in coated samples [33]. 

#### 3.2.7. Decay Incidence 

Percentage decay incidence was observed throughout the storage of 21 days at 8 °C. An increase in decay of the samples became more pronounced between 7–14 and 14–21 days interval. All the fruits in the control treatment showed microbial infection with different levels of severity on the disease incidence scale (Figure 4c). Maximum disease incidence of 67.33% was observed for uncoated samples during the first week of storage and corresponded with the results of Klangmuang and Sothornvit [37]. The decay incidence decreased with an increase in beeswax concentration, with S4 being the most efficient in reducing fruit decay. This may be attributed to the antimicrobial properties and inhibitory action of chitosan and *Aloe vera*, which helped to maintain the cellular integrity. Amino group of chitosan attaches to the cell membrane of bacteria through electrostatic forces, which permanently destroys the bacterial cell [10]. Additionally, *Aloe vera* extract used in coatings might have released phenolic compounds to thwart microbial growth [27]. Hinderance in the transport of oxygen from the environment to the surface of fruit might also have contributed to reduced microbial growth [9].

## 4. Conclusions

In this study, the potential of beeswax-emulsified chitosan–*Aloe vera* coatings was explored for the preservation of mangoes. Increased concentration of beeswax (0.5 to 2%) in coating formulations produced significantly stable emulsions with decreased polydispersity and average particle size of 4 µm. Hydrophobicity of beeswax and better cross-linkages between polymer and bioactive compounds in *Aloe vera* extract reduced the water transmission and diffusion coefficient to 33.83% and 43.78%, respectively. Furthermore, coatings having variable beeswax concentration significantly conserved the physicochemical characteristics of fruits, i.e., weight loss, firmness, pH, and total soluble solid contents. Coatings also reduced the oxygen permeability, which minimized the oxidation of phenolic compounds in the coated fruits. A significant reduction in decay incidence (4.2% on coated fruits with S4 formulation as compared to 62.1% in uncoated fruits) reveals their potential for practical application to decrease senescence and ripening of fruits during their exposure to market conditions.

## Figures and Tables

**Figure 1 foods-10-02240-f001:**
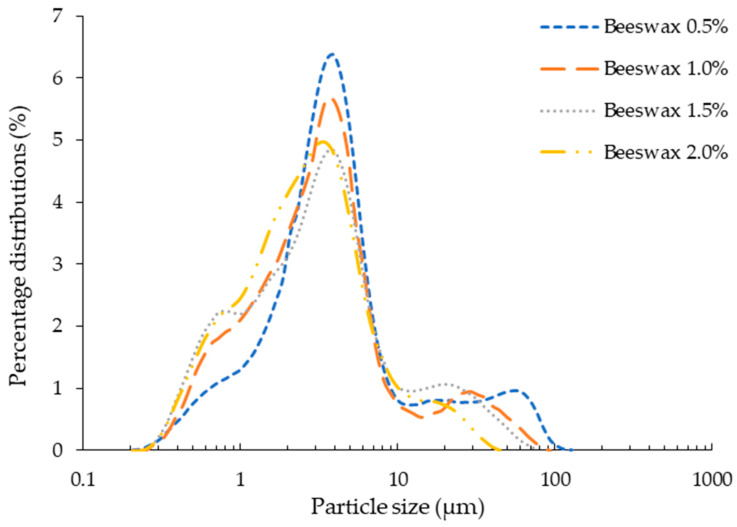
Particle size distribution of emulsion.

**Figure 2 foods-10-02240-f002:**
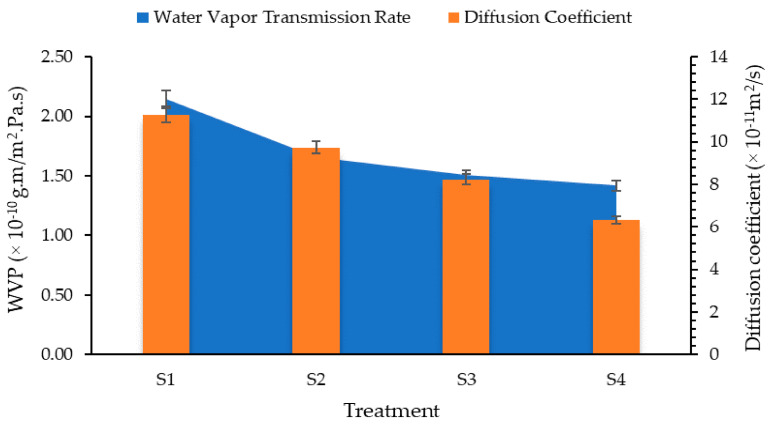
Water vapor permeability and diffusion coefficient of chitosan–*Aloe vera* gel coatings with different concentrations of beeswax.

**Figure 3 foods-10-02240-f003:**
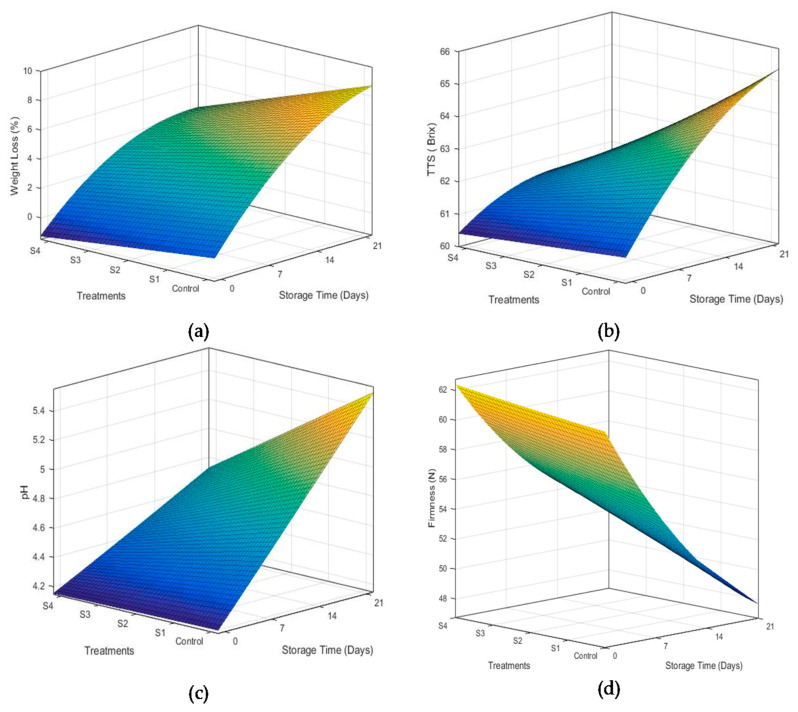
Effect of storage time and beeswax concentration on the various properties of coated mangoes, i.e., weight loss (**a**), total soluble solids (**b**), pH (**c**) and firmness (**d**) during storage at 18 °C and 75 ± 5% R.H.

**Figure 4 foods-10-02240-f004:**
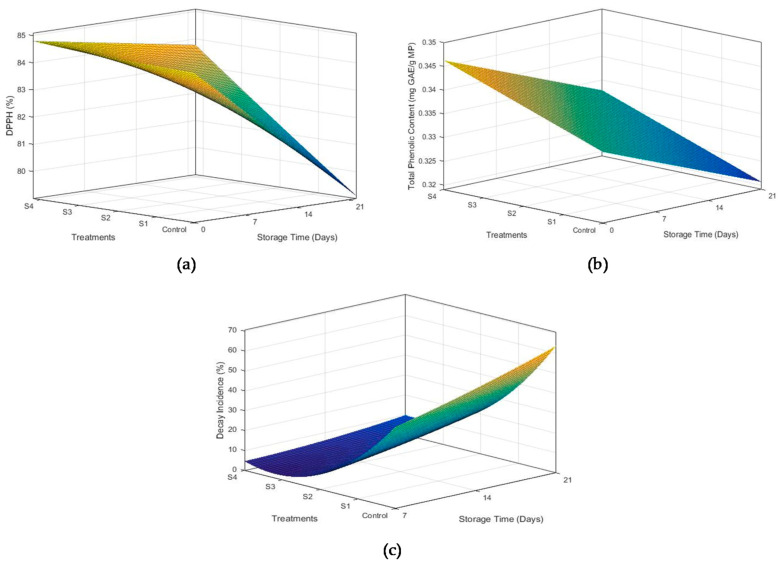
Effect of time and beeswax concentration on DPPH activity (**a**), total phenolic contents (**b**), and decay incidence (**c**) of mango fruits stored for 21 days at 18 °C and 75 ± 5% R.H.

**Table 1 foods-10-02240-t001:** Proportions of the emulsions used to formulate the different treatments.

Treatment	*Aloe vera*—Chitosan Solution (mL)	Beeswax (Melted) (mL)
S1	99.5	0.5
S2	99.0	1.0
S3	98.5	1.5
S4	98.0	2.0

## Data Availability

The data presented in this study are available on request from the corresponding authors.

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
