# Peer review of "Improvement of the Performance of Chitosan—Aloe vera Coatings by Adding Beeswax on Postharvest Quality of Mango Fruit"

_foods, 2021, doi:10.3390/foods10102240_

Round 1
Reviewer 1 Report
The authors investigated the effect of chitosan-Aloe vera bee wax emulsion on mango fruit in terms of preservation and improvement of shelf-life. The aim of the study was interesting, but some results should be updated for better understanding and quality.
Some critical suggestions
- The keyword “functional” is not appropriate. Please revise it.
- Why the mangoes were preserved at 18° Are there any specific reasons? Please discuss.
- The authors suggested including the chitosan nanoemulsion coating in their introduction. https://doi.org/10.1016/j.porgcoat.2020.106010.
- Surprised to see the particle size distribution results that how the higher concentration of bee wax decreased the particles size? Please explain.
- The author developed the nanoemulsion and film using chitosan-Aloe vera, and bee wax. Why was this study not focused on morphological and functional characterization? For example, Photographical evidence for film, AFM, FTIR.
- The authors suggested providing the experimental evidence for mangoes preservation at a different time interval. At least, provide the initial, and final day of mangoes appearance.
- Figure 1. legend “droplets” should be removed.
- Section 3.2.3. pH. The first statement is not correlated with the following statement. The authors suggested revising that section according to their results.
- The authors suggested checking their English language. For example, the authors mentioned in the results section “has been presented” but it should be “was presented”.
Author Response
Revision notes (response to reviewers)
Manuscript reference: foods-1368498
Manuscript title: Effect of chitosan-Aloe vera emulsion coatings on postharvest decay of mango (Mangifera indica)
Our response to the comments of the reviewer #1 is written in blue.
Referee(s)' Comments to Author:
Reviewer 1
Comments and Suggestions for Authors
The authors investigated the effect of chitosan-Aloe vera bee wax emulsion on mango fruit in terms of preservation and improvement of shelf-life. The aim of the study was interesting, but some results should be updated for better understanding and quality.
Some critical suggestions
The keyword “functional” is not appropriate. Please revise it.
Answer: Thank you for your suggestion. This keyword has been changed by “bioactive”.
Why the mangoes were preserved at 18° Are there any specific reasons? Please discuss.
Answer: The purpose of the study was to find a method to increase the shelf life of the product during marketing. The study was designed during the earlier period of Covid-19 to counter the deterioration of fruits while placed in the market. So, 18 °C was chosen to mimic the marketing conditions. This fact is also explained in “Materials and methods”
The authors suggested including the chitosan nanoemulsion coating in their introduction. https://doi.org/10.1016/j.porgcoat.2020.106010.
Answer: The reference has been included in the introduction.
Surprised to see the particle size distribution results that how the higher concentration of bee wax decreased the particles size? Please explain.
Answer: In general, the concentration of oil does not affect the average particle size as long as the concentration is below some critical value. In our case, the concentration of bee wax (i.e., up to 2%) was not that high and therefore no significant effects were observed on average particle size as is also reported in respective section (average particle size in all cases was around 4 mm). However, slight increase in polydispersity was observed. In some of the articles, a slight increase in size is reported with increasing oil concentration while in some others an opposite trend has also been reported. Actually, this is a complex phenomenon and depends on several factors such as type and concentration of oil, type and concentration of surfactants/stabilizer etc. An example can be seen in following article:
Hadnadev, T.D.; Dokic, P.; Krstonosic, V.; Hadnadev, M. Influence of oil phase concentration on droplet size distribution and stability of oil-in-water emulsions. Eur. J. Lipid Sci. Technol. 2013, 115, 313-321.
In another article (below), the particle size is related to the oil volume fraction in emulsion. It increases with increasing oil concentrations (at low oil volume fractions) and decreases when oil volume fraction exceeds some critical limit.
Binks, B.P.; Whitby, C.P. 2004. Silica particle stabilized emulsions of silicone oil and water: Aspects of emulsification. Langmuir 2004, 20, 1130-1137.
The author developed the nanoemulsion and film using chitosan-Aloe vera, and bee wax. Why was this study not focused on morphological and functional characterization? For example, Photographical evidence for film, AFM, FTIR.
Answer: The study was designed with the objective to increase the shelf life of mango fruits. Therefore, the study of characteristics of the coatings except a few related to stability, did not fit too well.
The authors suggested providing the experimental evidence for mangoes preservation at a different time interval. At least, provide the initial, and final day of mangoes appearance.
Answer: Photos were taken at the start of the experiment and with each interval till the end of study. Unfortunately, pictures are not available at this time due to some technical reasons.
Figure 1. legend “droplets” should be removed.
Answer: Done.
Section 3.2.3. pH. The first statement is not correlated with the following statement. The authors suggested revising that section according to their results.
Answer: Revised as suggested.
The authors suggested checking their English language. For example, the authors mentioned in the results section “has been presented” but it should be “was presented”.
Answer: English and grammar of the paper has been reviewed by an English speaker.

Reviewer 2 Report
In this research manuscript, authors developed an edible coating comprised of chitosan-Aloe vera coatings emulsified with beeswax for improving the storage stability of mangoes. This manuscript was described with detailed analysis. Following are my queries:
- Introduction looks simple. A short review on the edible coating may help the readers to understand further.
- Edible biofilms on Aloe vera-chitosan and beeswax was published by the same group in 2019. A brief of the edible films can be included in the introduction part.
- Section 2.3: Line 94 mentioned as the 20% Aloe vera-chitosan solution was prepared by mixing 20 ml of Aloe vera with 80 ml of prepared chitosan solution (1%). Similarly, in table 1, authors mentioned Aloe vera-chitosan solution (%) for different treatment ranging from 99.5 % to 98.0 %. Does the 99.5 % represent, 99.5 mL of aloe vera with 0.5 mL of 1 % chitosan solution? Does the high concentration of aloe vera affect the visual appearance of the final film?
- Section 2.5.4: Need to expand the keywords of equation 3 and 4.
- Line 140: What is “marked sample” represents?
- Line 144: “Conditions here” What are those conditions?
- Statistical analysis section should be described further. Description for considering the statistical significance is missing.
- Section 3.1.2. Standard deviation for the water vapor permeability and diffusion coefficient can be given. And the error bars can be included in Figure 2.
- Author can consider the sensory property in their future work and how the coating influences the taste of mangoes.
Author Response
Revision notes (response to reviewers)
Manuscript reference: foods-1368498
Manuscript title: Effect of chitosan-Aloe vera emulsion coatings on postharvest decay of mango (Mangifera indica)
Our response to the comments of the reviewer #2 is written in red.
Referee(s)' Comments to Author:
Reviewer 2
Comments and Suggestions for Authors
In this research manuscript, authors developed an edible coating comprised of chitosan-Aloe vera coatings emulsified with beeswax for improving the storage stability of mangoes. This manuscript was described with detailed analysis. Following are my queries:
Introduction looks simple. A short review on the edible coating may help the readers to understand further.
Answer: Thank you for your appreciation. The introduction has been revised and the suggestions proposed by the reviewer have been incorporated.
Edible biofilms on Aloe vera-chitosan and beeswax was published by the same group in 2019. A brief of the edible films can be included in the introduction part.
Answer: Details have been incorporated in the introduction, as well as a review of the literature.
Section 2.3: Line 94 mentioned as the 20% Aloe vera-chitosan solution was prepared by mixing 20 ml of Aloe vera with 80 ml of prepared chitosan solution (1%). Similarly, in table 1, authors mentioned Aloe vera-chitosan solution (%) for different treatment ranging from 99.5 % to 98.0 %. Does the 99.5 % represent, 99.5 mL of aloe vera with 0.5 mL of 1 % chitosan solution? Does the high concentration of aloe vera affect the visual appearance of the final film?
Answer: Yes, 99.5 and 5% represent mL. The correction has been made in the final draft. Moreover, the color of coating solution was dominated by the beeswax color, which was slightly yellowish in color.
Section 2.5.4: Need to expand the keywords of equation 3 and 4.
Answer: Done.
Line 140: What is “marked sample” represents?
Answer: All the samples were marked with specific keyword and a number (Controli, S1i, S2i, S3i, S4i; where i=1, 2, 3, 4, ... n) to better understand the samples. To avoid confusions, it has now been explained in the previous section (2.4. Application of coatings on mango fruits) of material and methods.
Line 144: “Conditions here” What are those conditions?
Answer: This was a typing error and it has been removed from the paragraph.
Statistical analysis section should be described further. Description for considering the statistical significance is missing.
Answer: Statistical analysis has been explained in detail.
Section 3.1.2. Standard deviation for the water vapor permeability and diffusion coefficient can be given. And the error bars can be included in Figure 2.
Answer: The figure has been improved according to the reviewer's suggestion
Author can consider the sensory property in their future work and how the coating influences the taste of mangoes.
Answer: Thank you very much for your suggestion, it will be taken into account for the design of future experiments.

Round 2
Reviewer 1 Report
The authors have improved the content and results of the manuscript in the revised version. Hence, the manuscript may be accepted for publication.
Author Response
Revision notes (response to reviewers)
Manuscript reference: foods-1368498
Manuscript title: Effect of chitosan-Aloe vera emulsion coatings on postharvest decay of mango (Mangifera indica)
Our response to the comments of the reviewer #1 is written in blue.
Referee(s)' Comments to Author:
Reviewer 1
Comments and Suggestions for Authors
The authors have improved the content and results of the manuscript in the revised version. Hence, the manuscript may be accepted for publication.
Answer: Thank you very much for your words and the effort made in reviewing the manuscript that has allowed us to improve it.

Reviewer 2 Report
Satisfied with the revised manuscript. Author incorporated the suggestions in the revised version.
Author Response
Revision notes (response to reviewers)
Manuscript reference: foods-1368498
Manuscript title: Effect of chitosan-Aloe vera emulsion coatings on postharvest decay of mango (Mangifera indica)
Our response to the comments of the reviewer #2 is written in green.
Referee(s)' Comments to Author:
Reviewer 2
Comments and Suggestions for Authors
Satisfied with the revised manuscript. Author incorporated the suggestions in the revised version.
Answer: Thank you very much for your words and the effort made in reviewing the manuscript that has allowed us to improve it.
